# Metabolomics Analysis of Sporulation-Associated Metabolites of *Metarhizium anisopliae* Based on Gas Chromatography–Mass Spectrometry

**DOI:** 10.3390/jof9101011

**Published:** 2023-10-13

**Authors:** Hua Yang, Longyan Tian, Hualong Qiu, Changsheng Qin, Siquan Ling, Jinzhu Xu

**Affiliations:** Guangdong Provincial Key Laboratory of Silviculture Protection and Utilization, Guangdong Academy of Forestry, Guangzhou 510520, China; yanghua@sinogaf.cn (H.Y.); tianlongyan@sinogaf.cn (L.T.); qiuhl@sinogaf.cn (H.Q.); qincs@sinogaf.cn (C.Q.); lingsiquan@126.com (S.L.)

**Keywords:** *Metarhizium anisopliae*, metabolomics, sporulation

## Abstract

*Metarhizium anisopliae*, an entomopathogenic fungus, has been widely used for the control of agricultural and forestry pests. However, sporulation degeneration occurs frequently during the process of successive culture, and we currently lack a clear understanding of the underlying mechanisms. In this study, the metabolic profiles of *M. anisopliae* were comparatively analyzed based on the metabolomics approach of gas chromatography–mass spectrometry (GC–MS). A total of 74 metabolites were detected in both normal and degenerate strains, with 40 differential metabolites contributing significantly to the model. Principal component analysis (PCA) and potential structure discriminant analysis (PLS-DA) showed a clear distinction between the sporulation of normal strains and degenerate strains. Specifically, 23 metabolites were down-regulated and 17 metabolites were up-regulated in degenerate strains compared to normal strains. The KEGG enrichment analysis identified 47 significant pathways. Among them, the alanine, aspartate and glutamate metabolic pathways and the glycine, serine and threonine metabolism had the most significant effects on sporulation, which revealed that significant changes occur in the metabolic phenotypes of strains during sporulation and degeneration processes. Furthermore, our subsequent experiments have substantiated that the addition of amino acids could improve *M. anisopliae*’s spore production. Our study shows that metabolites, especially amino acids, which are significantly up-regulated or down-regulated during the sporulation and degeneration of *M. anisopliae*, may be involved in the sporulation process of *M. anisopliae*, and amino acid metabolism (especially glutamate, aspartate, serine, glycine, arginine and leucine) may be an important part of the sporulation mechanism of *M. anisopliae.* This study provides a foundation and technical support for rejuvenation and production improvement strategies for *M. anisopliae*.

## 1. Introduction

The entomopathogenic fungus *Metarhizium* is extensively utilized for the control of agricultural and forestry pests, exhibiting efficacy against over 300 target pest species [1,2]. Consequently, there is significant international recognition and emphasis on the research and development of biopesticides, incorporating highly virulent strains of *Metarhizium* as their primary component [3]. However, during the industrialized production and preservation of germplasm resources, *Metarhizium* has undergone degeneration after multiple successive cultures, resulting in a significant decrease in conidia production, virulence reduction and loss of application value [4]. Strain degeneration has long been a challenge for the commercial production and application of entomopathogenic fungi. *Metarhizium* reproduces asexually to generate conidia. Sequential cultivation of conidia or mycelium is essential in the production of *Metarhizium* insecticides; however, this process often leads to strain degeneration. There is substantial variance between strains in terms of the impact of in vitro successive culture on the virulence and morphological traits of entomopathogenic fungi [5]. Germination rate, spore size and epidermal adhesion are the primary factors affecting the pathogenicity of pathogenic fungi [6]. Besides, bioassay, spore production on artificial media, optimal growth temperature, resistance to solar ultraviolet rays and toxin production have been considered as important parameters for screening highly virulent strains [7]. For example, the virulence of *Beauveria bassiana* strains is impacted by nutrition, and lower C/N ratios of artificial media tend towards greater virulence [8]. The activity of glutathione peroxidases and the scavenging ability of cellular reactive oxygen species (ROS) are improved in stabilized strains of *B. bassiana*. Overexpression of glutathione peroxidases, furthermore, can restore the reproductive capability of *Cordyceps militaris* [9]. Meanwhile, the decline in conidial production and virulence following sequential cultivation is frequently accompanied by alterations in secondary metabolites [10]. When comparing differentially methylated regions of genomic DNA in wild-type and mutant mycelia of *C. militaris*, it was discovered that the pathways associated with glycerophospholipid metabolism, pyruvate metabolism, ubiquitin-mediated protein hydrolysis and n-glycan biosynthesis may be linked to the spore-producing degradation [11]. The level of activity observed in laccase, carboxylester hydrolase, α-galactosidase and catalase in flat mushrooms presented a direct correlation to the number of passages, which could be used as an indicative characteristic of strain degeneration [12]. Therefore, the degradation of the strain can be attributed to either inhibition or synergistic effects on the synthesis of metabolites involved in metabolic regulation.

Metabolomics, as a component of systems biology, is dedicated to elucidating the metabolic reactions induced by environmental or genetic factors through the analysis of changes in small molecule metabolites, and subsequently unraveling complex physiological mechanisms. Currently, metabolomics has diverse applications, including cellular phenotype classification and gene function, as well as metabolism studies on transformed bacterial strains. Microorganisms are widely utilized in metabolomics analysis and research due to their singular structure, uncomplicated nutritional requirements and well-developed enzyme systems. Numerous microorganisms, including *Saccharomyces cerevisiae* and *Escherichia coli*, possess significant medical, pharmacological and industrial production values. Mashego et al. [13] explored the effects of culture age on the levels of intracellular and extracellular metabolites, as well as the specific activities of enzymes involved in central carbon metabolism in *S. cerevisiae* over 90 generations. They found that there was a down-regulation of metabolic overcapacities (for reversible reactions) and storage pools (such as trehalose, amino acids and excess metabolic capacity in glycolytic protein in enzymes) over the course of evolution. Karpe et al. [14] examined the sugar metabolism of four types of filamentous fungi using gas chromatography–mass spectrometry (GC–MS). They found that the major metabolites generated during fungal degradation, which differentiated the metabolic profiles of fungi, included sugars, sugar acids, organic acids and fatty acids. Eshell et al. [15] found that the degradation of Aflatoxin B1 of Actinomycetes was associated with the appearance of a range of lower molecular weight compounds and coupled with the accumulation of intermediates of fatty acid metabolism and glycolysis. Metabolomic analysis in combination with insect tissue culturing shows that two generalist isolates of the genus *Metarhizium* and *Beauveria* employ significantly different arrays of secondary metabolites during infectious processes and saprophytic growth [16]. During the transition from mycelia to fruiting bodies of *Cordyceps bassiana*, the major metabolic change observed was the conversion of glucose to mannitol, and beauvericin to phenylalanine and 1-hydroxyisovaleric acid [17]. In recent years, the rapid advancement of multi-omics and the utilization of bioinformatics have facilitated the discovery of a greater number of regulatory genes, unveiling crucial synthesis pathways for various metabolites [18,19,20,21]. Therefore, investigating metabolite biosynthesis and functional genes holds significant practical value in elucidating the degenerate mechanisms of *Metarhizium*.

In this study, we utilized the GC-MS based metabolomics sample processing method established by our group for *Metarhizium anisopliae* [22] to investigate the correlation between secondary metabolite metabolism and sporulation degeneration in different strains (Figure 1) of *M. anisopliae*. By employing chromatography–mass spectrometry techniques, we systematically analyzed the metabolic differences between normal and degenerate strains of *M. anisopliae* preserved in our laboratory. By comparing the metabolome differences between normal and degenerate strains, we searched for endogenous small molecule biomarkers and metabolic pathways closely related to the sporulation of *M. anisopliae*, elucidating the relationship between the degeneration of *M. anisopliae* and the metabolism of secondary substances. The aforementioned elucidation of the relationship between *M. anisopliae* degeneration and secondary metabolism has laid a solid foundation for subsequent comprehension of metabolome regulation in conjunction with transcriptome and proteome regulation, as well as entomopathogenic fungi degeneration. Further studies in this direction will yield crucial biological insights for the rejuvenation and preservation of *M. anisopliae*, thereby facilitating the industrialization of entomopathogenic fungi and promoting sustainable agroforestry development.

## 2. Materials and Methods

### 2.1. Fungi Materials

The strain *M. anisopliae* 09 (*Ma*09), which exhibited superior sporulation, was selected for multiple spot grafting and prolonged cultivation (for 25 generations) until a significant decline in sporulation was observed. The normal strains (the first generation strains) and the degenerate strains (the 25 generations) (Figure 1) were then inoculated onto a fresh PPDA plate and incubated at 25 °C for seven days to produce conidia. A plug of fungal agar culture was prepared by punching out a 6 mm circle along the outer edge of the colony and was delicately placed onto a solid PPDA medium dish that had been coated with cellophane. It would be cultivated further under controlled conditions at 25 °C over the course of five to seven days. Subsequently, the mycelial growth on the surface of the cellophane layer would be scrupulously excised using disinfected scalpels. The sporulation quantity of normal strains and degenerate strains was determined to be 10.28 and 1.23 × 10^7^ CFU, respectively (Figure 1). After collecting approximately (100 ± 5) mg of this tissue sample, it would then be promptly flash-frozen in liquid nitrogen for three to five minutes as an effective means of deactivating any residual biological activity prior to storage at −80 °C.

### 2.2. Metabolome Analysis

Extraction of metabolites: The preserved mycelium samples were soaked in pre-cooled methanol (−80 °C), homogenized using a ball mill at a frequency of 50 times/s for one minute, subjected to vortex oscillation for one minute, and then underwent ultrasonic extraction lasting twenty minutes. The resulting supernatant was transferred into another centrifuge tube where it received an additional 1 ml volume of extraction solution; this process was repeated twice prior to combining the supernatants, which were subsequently dried under nitrogen gas before undergoing derivatization treatment.

Derivation: After lyophilization, the extracted samples were treated with 150 μL of a pyridine solution containing 20 mg/mL methoxyamine hydrochloride. The mixture was vigorously oscillated for thirty seconds and then incubated at 37 °C for ninety minutes. After cooling to room temperature, a BSTFA derivative containing 1% TMCS (50 μL) was added and the reaction proceeded at 70 °C for one hour. Following a standing period of thirty minutes at room temperature, GC–MS analysis was performed, or the sample was stored at −20 °C.

GC-MS conditions: The inlet temperature was set to 280 °C, and the heating procedure involved an initial temperature of 50 °C for three minutes, followed by a heating rate of 10 °C/min to achieve a temperature of 150 °C within a period of five minutes. Subsequently, there was a gradual increase in temperature at a rate of 5 °C/min until it reached 200 °C within another span of five minutes. Finally, there was an abrupt rise in temperature at a rate of 10 °C/min to reach the final target temperature of 280 °C during the last five minute interval. High purity helium gas (purity ≥ 99.999%) was utilized as the carrier gas, employing a non-shunt injection method with an injection volume of 1 µL at a flow rate of 1 mL/min. The electron impact ionization source (EI) was employed, and the ion source temperature, quadrupole temperature and interface temperature were set to 230 °C, 150 °C and 280 °C respectively. The electronic energy was calibrated to 70 eV using standard tuning methods for mass spectrometry, scanning in full scan mode over a scanning range of 40–500 amu. Spectral retrieval was conducted utilizing the NIST (National Institute of Standards and Technology) 2017 spectral library.

Data pre-processing and recognition: Peaks with ratios (signal/noise) of <5 were removed. The original GC-MS data were automatically analyzed using the automatic mass spectral deconvolution and identification system (AMDIS) and identified by comparing to the database of NIST 2017. The material identification is based on comparing the EI mass spectrometry fragments of the detected metabolites to the mass spectrometry information of the standard substances stored. The NIST 2017 database retrieval matching criterion is that the similarity of the mass spectrometry fragments is greater than 700 (the highest value is 1000). The information related to the identified metabolites was collected from the spectra, including retention time, peak area and name of the substance [23].

### 2.3. Effects of Different Amino Acids on Sporulation

The 6 amino acids (Serine, L-alanine, Glutamate, Aspartate, Arginine, Leucine) detected in the experiment were selected to cultivate the degraded spore-producing Metarhizium anisopliae by plate grafting method. The effect of amino acid composition on sporulation was investigated based on the amount of sporulation. The 6 kinds of amino acids were configured in solution according to the following concentration: Aspartate 100 (mg/L), L-alanine 100 (mg/L), arginine 15 (mg/L), glutamate 140 (mg/L), serine 20 (mg/L), leucine 100 (mg/L). After 0.22 um filtration, the solution was added to PPDA medium to make a 9 cm amino acid medium for the test Petri dish. The degraded strain of sporulation was selected as the test strain, transferred to the medium containing amino acids and subcultured for 5 generations, and the number of spores produced in each generation was counted. PPDA medium without added amino acids was used as a control. Five replicates were performed for each treatment.

### 2.4. Data Analysis

For subsequent analysis, a substance should only be considered if it is present in at least 3 out of the 6 replicate samples. By default, missing values will be replaced by 1/5 of the min positive values of their corresponding variables. We normalized each alignment using normalization by sum, then the data were log10 transformed. The data scaling was performed through auto scaling (the data were mean-centered and divided by the standard deviation of each variable). The resulting data matrix of peak area was then imported into SIMCA-P software (version 14.0, Umetrics, Umea, Sweden) for multivariate statistical analysis. Principal component analysis (PCA) and partial least squares-discriminant analysis (PLS-DA) (n = 6) were conducted on GC–MS data from normal strains and degenerate strains. Two-dimensional maps were generated to visualize the results of PCA and PLS-DA analyses. According to the PLS-DA model, the R2Y and Q2 of the permutation test were used to assess the goodness-of-fit and predictive ability of the PLS-DA models. The metabolites with VIP > 1 and *p*-value < 0.05 were considered to be differential metabolites. MetaboAnalyst 5.0 was adopted for the metabolic pathway analysis [24].

## 3. Results

### 3.1. Comparison of the Metabolism Difference of Ma09

The total ion chromatograms of the normal and degenerate strains are presented in Figure 2, with the former representing normal sporulation and the latter depicting a degenerate state. A total of 74 metabolic substances were detected. Evidently, significant metabolic disparities exist between these two strains, as well as shared endogenous metabolites. See Appendix A for detailed Data.

### 3.2. Multivariate Analysis of the Differential Metabolites

The PCA score plot displays the distribution of samples in the principal component space, with similar samples clustered together and those with differences distributed in distinct regions. As a result, metabolic differences can be classified based on the score chart. The results indicated that there is clustering of metabolites from the six replicates of each treatment source, and the fact that the two treatment ellipses do not overlap the first principal component explains 56.1% of the variance, whereas the second principal component explains 13.6% (Figure 3). Furthermore, the score plot exhibited a clear trend in distinguishing between normal and degenerate strains; specifically, the six treatments of normal strains are clustered on the left side of Figure 3, while the six treatments of degenerate strains were relatively dispersed on the right of Figure 3. This indicates that there exist significant inter-individual variations in GC–MS metabolism among degenerate strains. The load plot identifies differential variables that contribute to classification; each point on the graph represents a unique metabolic substance, and its contribution to the model is determined by its distance from the center point. The further away from the center point, the more significant its contribution to the model.

The PLS-DA method incorporates orthogonal signal correction (OSC) to eliminate signals that are not correlated with the model classification, thereby enhancing sample separation between groups and improving explanatory power. As demonstrated by the PLS-DA score plot (Figure 4), there is a clear distinction between the groups, characterized by a high degree of clustering, which underscores the reproducibility of this method. The first principal component explains 56% of the variance, whereas the second principal component explains 11.6%. The R2Y value of 0.996 and the Q2 value of 0.983 indicated the high reliability of the current model. To further assess its robustness, a permutation test was conducted. The *y*-axis intercept of Q2 was below 0.5, and when the transverse coordinate equaled 1, Q2 was less than R2 but very close to it, thus confirming the model’s reliability. Therefore, the original model adequately accounted for differences between samples.

The metabolites with VIP > 1.0 in PLS-DA analysis and *p* < 0.05 were used for further selecting the significantly differential metabolites by univariate analysis (Table 1). A total of 40 differential metabolites with VIP scores greater than 1 and *p* < 0.05 were identified as potential metabolic markers between the two groups. The levels of 23 substances were down-regulated, including 2,3-Diphosphoglyceric acid; Acetyl-CoA; Aflatoxin G2; Arginine; Asparagine; Carnitine; Ergothioneine; Galactitol; Glycine; Indomethacin; Leucine; L-Glutamate; L-Histidinol; Linolenic acid; L-Proline; 1-methyl-; methyl ester; Mannitol; Mesaconic acid; Octadecanoic acid; Phenylalanine; Pyruvate; Trehalose; Tyrosine; and Valine. Meanwhile, the levels of 17 substances were up-regulated, which include 9,10-Epoxyoctadecanoic acid-2-ethylhexyl ester; Aspartate; Betaine; Glutamine; Glyoxylate; Guggulsterone; L-Aspartyl-L-phenylalanine; Linoleic acid; Maleamic acid; Mandelic acid; Myo-inositol; N-Acetyl-L-phenylalanine; Oxoglutaric acid; Proline; Serine; Threonine; and Uracil. Hierarchical clustering analysis was performed on 40 metabolites to identify inter-component change patterns. Visual inspection aided in distinguishing and addressing differences between data sets. The heat map shows a clear difference between the normal and degenerate strains of *Ma*09 (Figure 5). The heat map facilitated simultaneous clustering of both samples and metabolites, grouping together those with similar metabolic profiles and aggregating metabolites related to metabolism. On the horizontal axis of the figure, normal and degenerate strains are clearly distinguished as two categories. On the vertical axis of the graph, metabolites related to biochemistry aggregate into one category.

### 3.3. Pathway Analysis

To gain insight into the metabolic pathways associated with sporulation, we conducted pathway enrichment and topology analysis based on metabolites exhibiting significant differences between normal and degenerate strains. Pathway enrichment analysis and path topology analysis were conducted on the two datasets, respectively. The results of metabolic pathway analysis are shown by a bubble chart. Each bubble represents a metabolic pathway in the bubble diagram. The abscissa and bubble size represent the influence factor of the path in topological analysis. The y-coordinate and bubble color represent the *p* values of the enrichment analysis. Through KEGG search for differential metabolites, a total of 47 metabolic pathways (Appendix A pathway_results.xls) were found to be involved. Figure 6 demonstrates the significant impact of the top 25 pathways on sporulation. Among the 47 metabolic pathways, there are 12 metabolic pathways related to biosynthesis; 28 metabolic pathways related to metabolism; 4 metabolic pathways related to degradation; and 3 other pathways. Additionally, there exist 9 amino acid-related pathways in the 47 metabolic pathways. Among them, our focus lies on investigating the influence of amino acids on sporulation. Based on our findings, the first two amino acid metabolic pathways that exert a significant impact on sporulation are alanine, aspartate and glutamate metabolism, along with glycine, serine and threonine metabolism.

Figure 7 and Figure 8 illustrate the two metabolic pathways that have been identified as having the most significant impact based on KEGG differential metabolites analysis. By integrating metabolic pathways with differential metabolites, a total of 16 biomarkers were identified in the metabolic network, including arginine, asparagine, aspartate, fumarate, N-(L-arginino)succinate, glutamate, glutamine, glycine, glyoxylate, 2-Phospho-D-glycerate, proline, pyruvate, serine, succinate, threonine, trehalose and tryptophan.

### 3.4. Amino Acids Have an Effect on Sporulation

The low spore-producing *Ma*09 was continuously cultured up to the fifth generation by supplementing the medium with various amino acids, and the corresponding changes in spore production are illustrated in Figure 9. The spore production of the first generation strain was determined to be 5.42 × 10^6^ CFU. The results demonstrated a pattern of increasing, followed by decreasing, and then increasing spore production when the medium contained serine and L-alanine. Additionally, an increase in spore production was observed when the medium contained glutamate, aspartic acid and arginine. Furthermore, an increase in spore production was noted with the inclusion of leucine in the medium; however, a decrease in spore production was observed when leucine was present. The spore production of *M. anisopliae* was found to be significantly higher in the medium supplemented with amino acids compared to the medium without amino acid supplementation, after four generations of *successive culture*. The spore production exhibited a declining trend in the absence of amino acid supplementation in the medium, suggesting that the addition of amino acids exerts a discernible impact on *M. anisopliae*’s spore production.

## 4. Discussion

Amino acids are crucial organic compounds that play a vital role in numerous biological processes, including protein synthesis, cell growth and development, and energy production. Research has shown that amino acids not only serve as key players in cell signal transduction, but also exert significant regulatory effects on gene expression and protein phosphorylation cascades [25].

Glutamate serves as a precursor for the biosynthesis of purine and pyrimidine nucleotide bases in organisms, while also playing a crucial role in various metabolic pathways as an amino group donor. Glutamate and glutamine exert a crucial regulatory function in the biosynthesis of glutathione. The catabolism of glutamine via the glutaminolysis pathway generates not only glutamate but also minor amounts of aspartate, alanine, lactic acid and pyruvate, while serving as a substrate for GABA synthesis [26]. The study revealed a decrease in glutamine content, indicating a reduction in the substance’s strength. Elevated levels of aspartate and argininosuccinic acid were observed, while the corresponding fumarate level was reduced, suggesting potential issues with the conversion process from aspartate to fumarate. The conversion of aspartate to fumarate can occur through two pathways. The first pathway involves the intermediate compound aspartate–adenylosuccinate–fumarate. In this experiment, adenylosuccinate was not detected, suggesting that the conversion of aspartate to adenylosuccinate was inhibited and, consequently, adenylosuccinate production was suppressed, leading to down-regulation of fumarate levels. Alternatively, it is possible that technical issues prevented the detection of adenylosuccinate. Therefore, the down-regulation of fumarate may also be attributed to the fact that certain genes have a part missing, or certain proteins are abnormally expressed in the second pathway involving aspartate–argininosuccinate–fumarate. The enrichment of argininosuccinate during its conversion to fumarate is attributed to genes missing or protein expression inhibition, leading to up-regulation. Inconsistent material change trends were observed in the conversion processes of aspartate–aspartamide and glutamate–glutamine before and after, suggesting issues in these pathways resulting in lower conversion rates (Figure 10). Aspartate and glutamine also have a certain effect on subsequent arginine and proline metabolism, and L-arginine is necessary for the sporulation of biocontrol fungus *Coniothyrium minitans*, plus, its derivative nitric oxide may mediate its sporulation function [27]. Proline plays a crucial role in stabilizing the structure and function of collagen in vivo, safeguarding the integrity of the central nervous system, regulating cell osmotic pressure, as well as enhancing salt and drought tolerance [28]. In the metabolic pathway of glycine, serine and threonine, this experiment also found that glycine produced a weakening and up-regulation of glyoxylate, threonine and serine, suggesting that the conversions behind these three substances were inhibited, which caused the accumulation of upstream substances. The remarkable incorporation of glycine and serine into the wild-type strain during sporulation was closely associated with spore coat formation. Higher levels of glycine may be a response to high temperature stress, which is closely associated with spore and conidium production [29].

Other metabolites such as oleic, linoleic and linolenic acids have also been reported to be important sources of oxylipins in pathogenic fungi. As a vital signaling molecule, oxylipins play a pivotal role in fungal growth; sexual and clonal reproduction; host–fungal signal recognition; host immune defense; and pathogenicity [30,31,32]. Furthermore, the metabolism of oxylipins is also associated with mycotoxin synthesis. Trehalose, a non-reducing disaccharide widely distributed in the hyphae and spores of fungi, is typically induced under conditions of heat, desiccation and oxidative stress. Its primary function lies in maintaining cell membrane fluidity and protein stability [33]. In fungi, mannitol can undergo conversion into mannitol 1-phosphate through the action of enzymes such as mannitol kinase and mannitol 1-phosphate dehydrogenase. This process subsequently generates fructose-6-phosphate which enters the glycolytic cycle, thereby participating in energy metabolism [34]. Mannitol plays a crucial role in maintaining cellular osmotic pressure, fungal stress resistance and host interaction. Mannitol also plays an essential role in asexual sporulation in *Stagonospora nodorum* both in vitro and in planta [35]. Furthermore, the pathogenicity of pathogenic fungi can also be influenced by mannitol [36,37].

The majority of organic materials undergo oxidation and degeneration during the tricarboxylic acid cycle [38]. Inhibiting this cycle can effectively reduce excessive consumption of organic materials, thereby promoting spore biosynthesis. Two metabolic pathways—the alanine, aspartate and glutamate metabolic pathway and the glycine, serine and threonine metabolism—had the most significant effects on sporulation, mainly related to unsaturated fatty acid synthesis, lysosomes and amino acid metabolism. In addition, we identified other kinds of compounds, such as amides, that can be used in the formation of secondary metabolites, which play a key role in spore formation. Significant changes in the levels of major metabolites may indicate a key role in multiple metabolic pathways that regulate the sporulation of *M. anisopliae*. Using GC–MS, several hundred compounds could be analyzed simultaneously, including organic acids, most amino acids, sugars, sugar alcohols, aromatic amines and fatty acids. But the metabolic differences in this study still have some limitations, and due to the limitations [39] of GC–MS, an incomplete range of metabolites was identified; in particular, lipids were less frequently identified, which may be due to their low volatility, resulting in the omission of metabolites. The derivatization reaction is an essential prerequisite in GC–MS analysis. However, the use of standardizations or standard operating procedures (SOPs) is insufficient to provide reliable results in metabolomic studies using derivatization-based GC–MS. Conversely, they may create a false sense of reliability. The derivatization procedure and origins of chromatographic peaks constitute two fundamental issues that can directly impact the outcome of GC–MS analysis [40,41]. Therefore, more sensitive high-throughput and advanced histologic techniques should be used to interpret the entire metabolic network.

## 5. Conclusions

At present, we have employed GC-MS metabolomics to elucidate the metabolic differences between strains exhibiting normal sporulation and those displaying impaired sporulation, as metabolites serve as a reliable indicator of physiological responses to multiple factors. So far, this is the first time the metabolic changes of spore-producing *Metarhizium* have been evaluated. A total of 74 metabolites were detected through rapid GC-MS analysis of normal and degenerate strains, with 40 differential metabolites identified as potential metabolic markers between the two groups. Enrichment analysis was conducted using KEGG, resulting in the identification of 47 relevant pathways. The KEGG pathway mapper and metabolite set enrichment analysis indicated that these metabolites are primarily associated with energy, amino acid synthesis and metabolism, and fatty acid synthesis and metabolism, among others. The amino acid metabolism (especially glutamate, aspartate, serine, glycine, arginine and leucine) may be an important part of the sporulation mechanism of *M. anisopliae.* The addition of amino acids to the medium can increase *M. anisopliae* spore yield.

## Figures and Tables

**Figure 1 jof-09-01011-f001:**
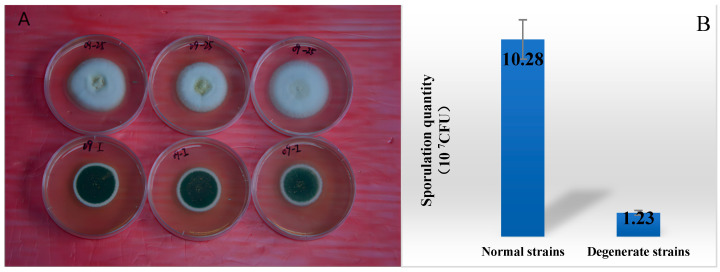
The colony morphology on the culture medium of *Ma*09 is shown. (**A**): The lower row displays the first generation of *Ma*09 strains with normal sporulation, while the upper row exhibits the degenerate strains after successive culture. (**B**): The sporulation quantity of normal strains was 10.28 × 10^7^ CFU; the sporulation quantity of degenerate strains was 1.23 × 10^7^ CFU.

**Figure 2 jof-09-01011-f002:**
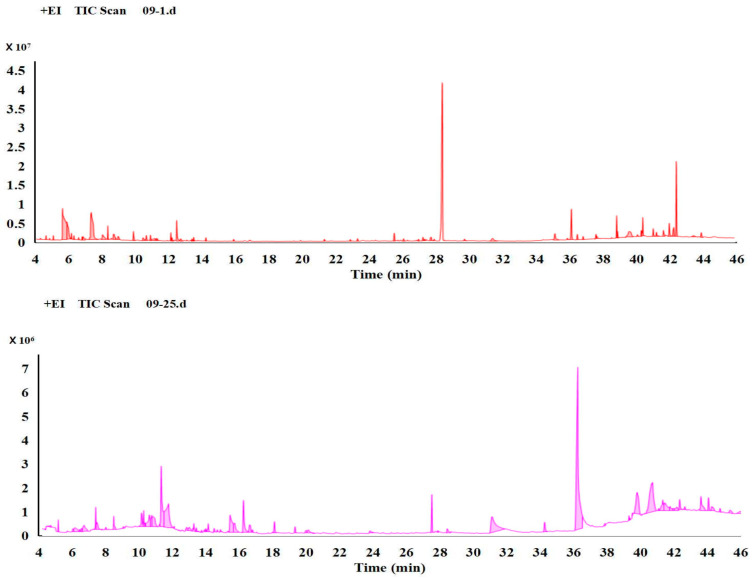
The typical total ion chromatograms (TICs) of the normal strains and their degenerate counterparts are presented. The *y*-axis represents relative mass abundance, while the *x*-axis denotes retention time. The top panel illustrates normal strains, whereas the bottom panel depicts degenerate strains.

**Figure 3 jof-09-01011-f003:**
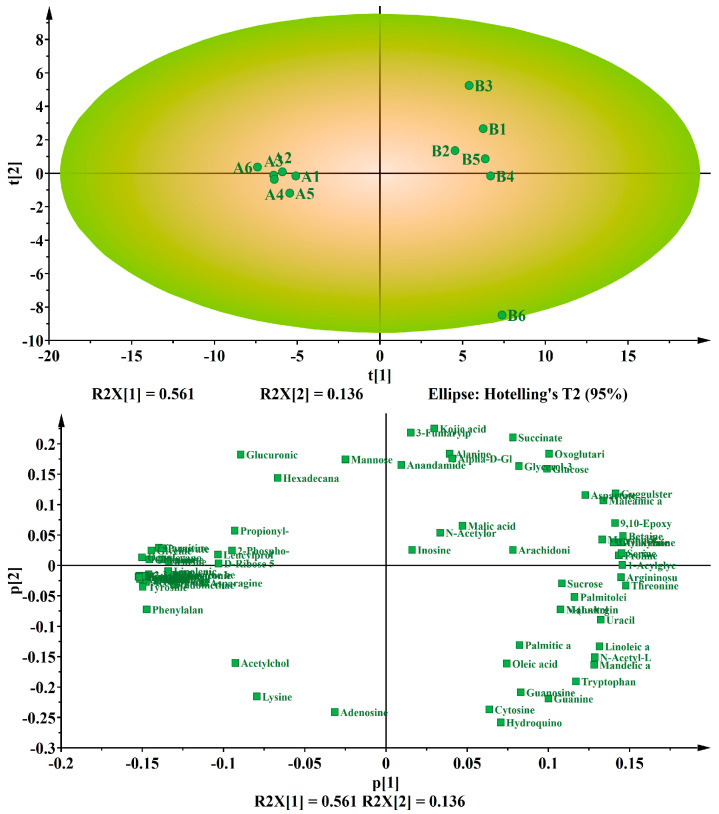
Principal component analysis (PCA) score plot (**up**) and loading plots (**down**) for separation of metabolites in normal strains and degenerate strains. The variances are shown in brackets. The six symbols for each treatment represent six biological replicates. Each green box represents a metabolite. A represents the normal strain and B represents the degenerate strains.

**Figure 4 jof-09-01011-f004:**
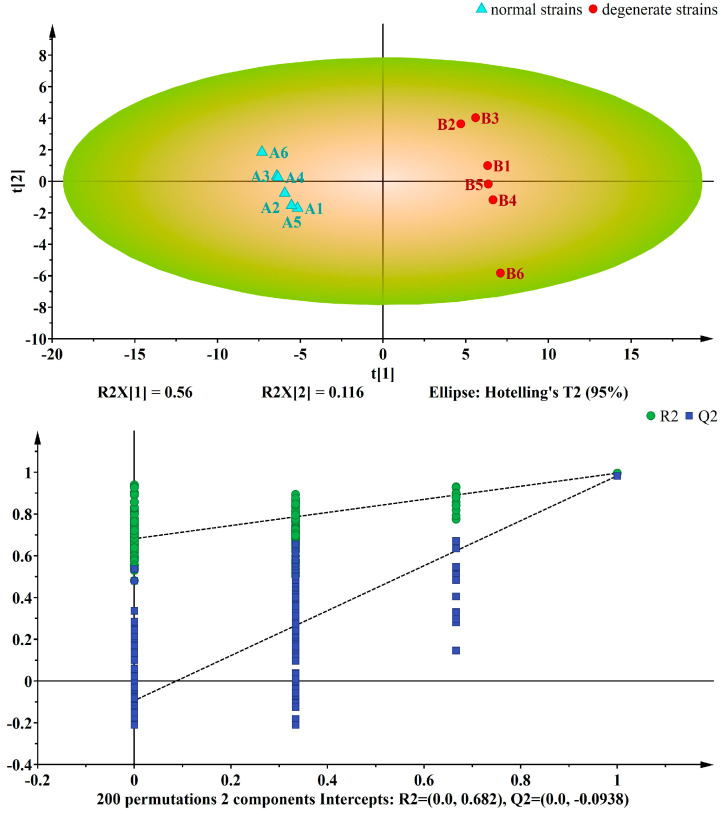
PLS-DA score plots (**up**) with corresponding permutation test plots (**down**) derived from GC–MS metabolite profiles of *Ma*09. The variances are shown in brackets. A represents the normal strains and B represents the degenerate strains.

**Figure 5 jof-09-01011-f005:**
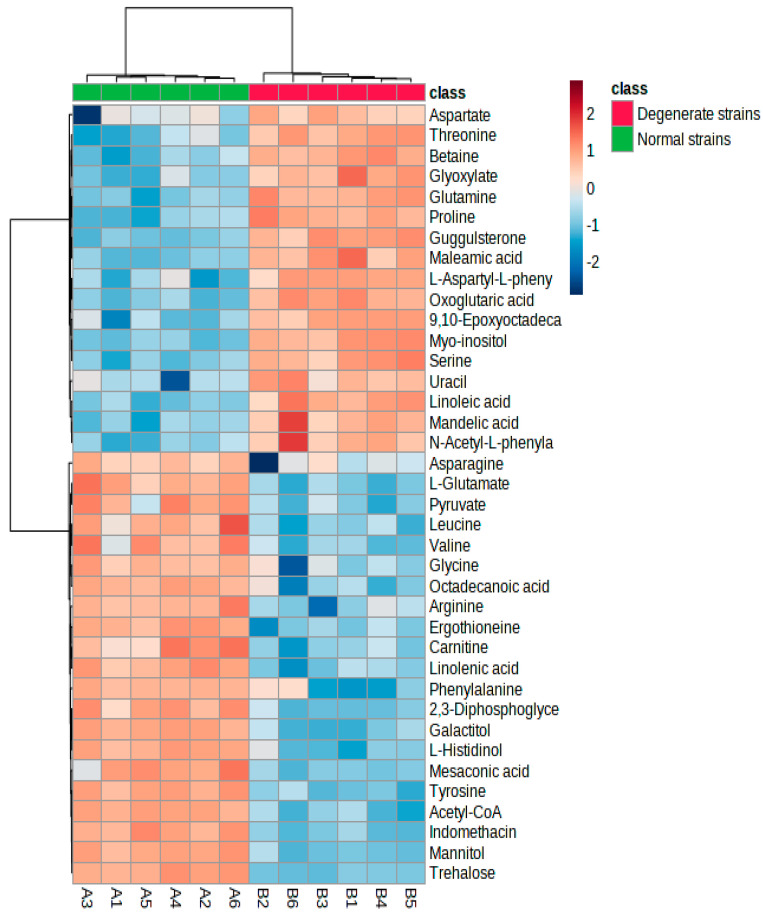
Heat map of metabolites obtained from normal strains in comparison to degenerate strains. The green columns represent 6 replicates of normal strains (A1–A6) and solid red represents 6 replicates of degenerate strains (B1–B6). Red indicates Z-scores > 0 and blue indicates Z-scores < 0. The saturation threshold is set at ±2 (Z-score −2 to +2 representing low to high values).

**Figure 6 jof-09-01011-f006:**
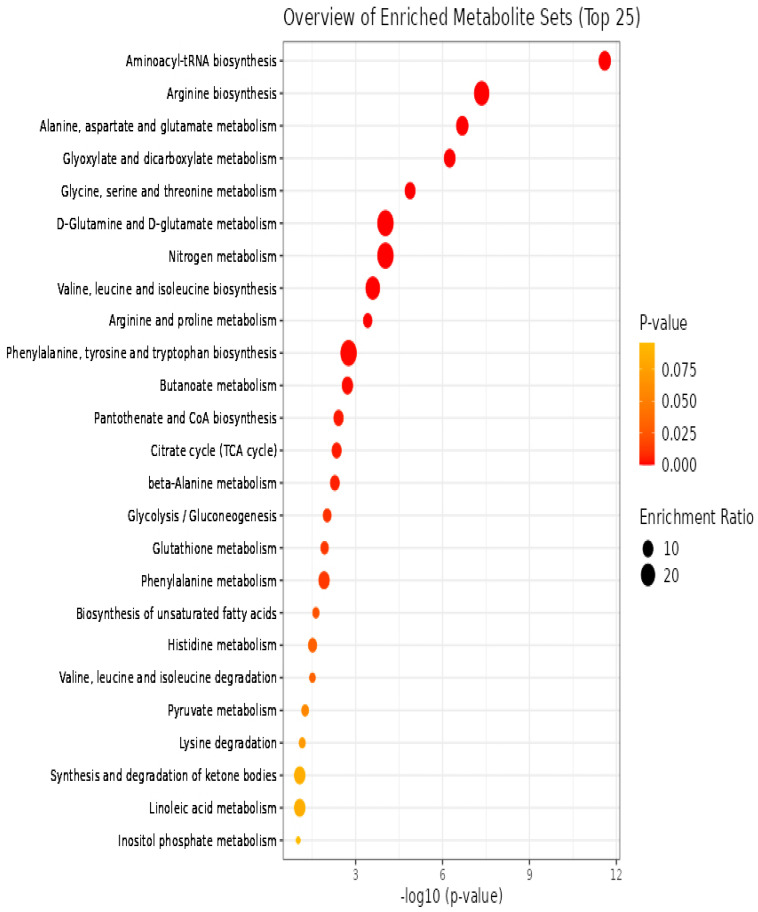
Metabolome map of significant metabolic pathways characterized in *Ma*09 for the normal and degenerate strains. Significantly changed pathways based on enrichment and topology analysis are shown. The *x*-axis represents pathway enrichment, whereas the *y*-axis represents the top 25 pathways. Large sizes and dark colors represent major pathway enrichment and high pathway impact values, respectively. The color ranges from orange to red, indicating that the *p*-value decreases successively. The larger the dot, the greater the number of metabolites enriched in the pathway.

**Figure 7 jof-09-01011-f007:**
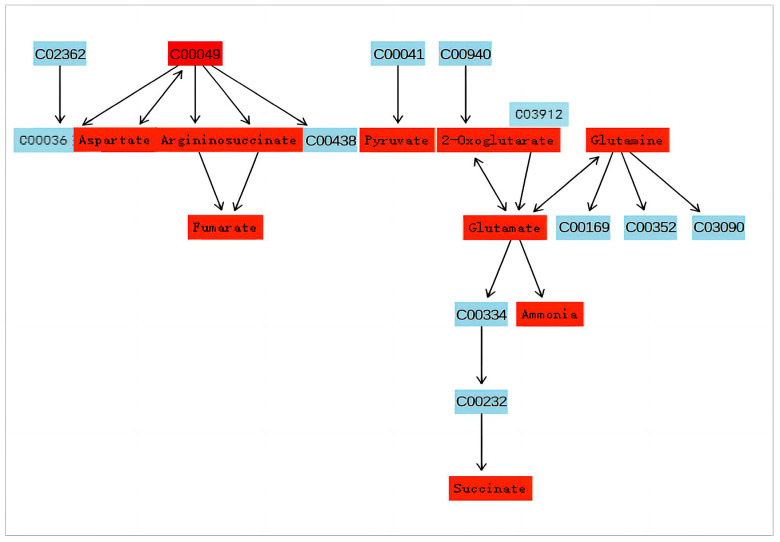
Alanine, aspartate and glutamate metabolism. The substance indicated in red is the one that has been detected on this occasion.

**Figure 8 jof-09-01011-f008:**
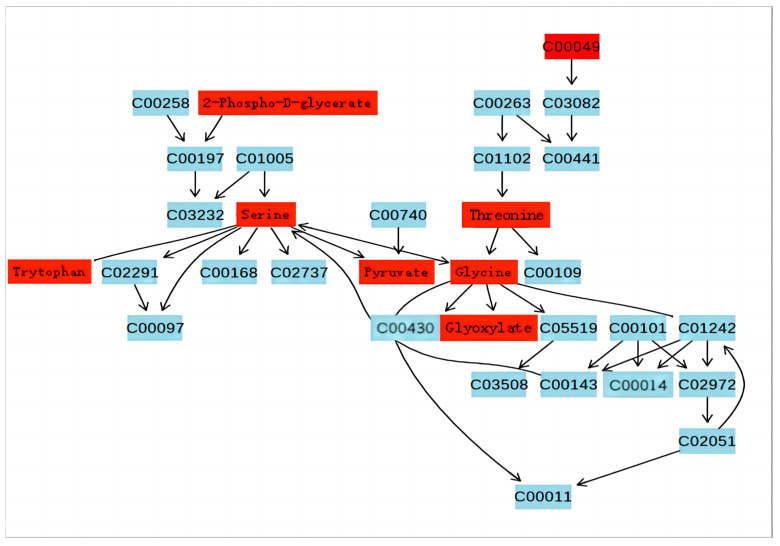
Glycine, serine and threonine metabolism. The substance indicated in red is the one that has been detected on this occasion.

**Figure 9 jof-09-01011-f009:**
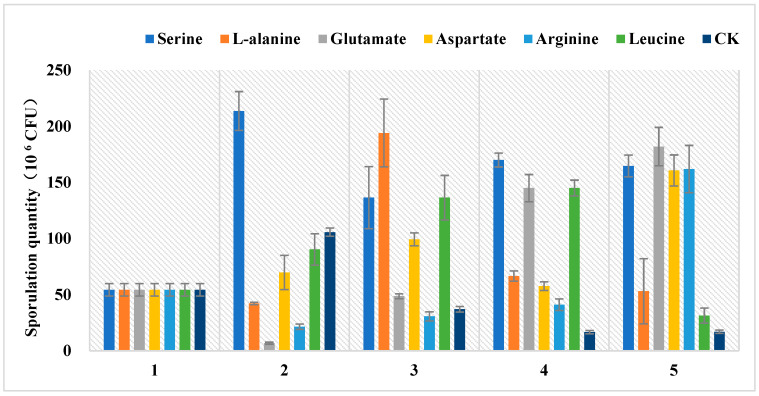
Effects of different amino acids on sporulation. The horizontal coordinate represents the number of generations of culture, the vertical axis represents sporulation quantity.

**Figure 10 jof-09-01011-f010:**
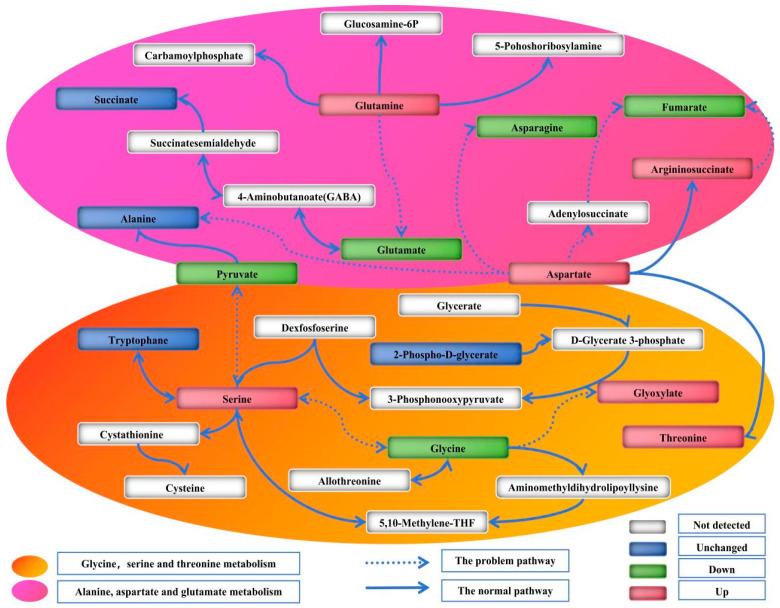
A working model of several major amino acid metabolic pathways. The white square indicates that the substance has not been detected; the green box in the figure indicates that the substance is down; the red box indicates that the substance is up; and the blue box indicates that the substance is unchanged. The orange area is the glycine, serine and threonine metabolism pathway, and the pink region is the alanine, aspartate and glutamate metabolism pathway.

**Table 1 jof-09-01011-t001:** Differential component information between normal strains and degenerate strains identified by GC–MS.

Components	*p*-Value	VIP	Trend
Acetyl-CoA	0	1.29195	down
Aflatoxin G2	0	1.17275	down
Ammonia	0	1.31058	down
Arginine	0	1.19034	down
Asparagine	0.005	1.01498	down
Aspartate	0.001	1.10915	up
Betaine	0	1.28929	up
Carnitine	0	1.16839	down
Ergothioneine	0	1.28954	down
Glutamate	0	1.24028	down
Glutamine	0	1.28597	up
1-Acylglycerophosphoinositol	0	1.24205	up
Glycine	0	1.20305	down
Glyoxylate	0	1.20335	up
Guggulsterone	0	1.24587	up
Histidinol	0	1.29683	down
Leucine	0.001	1.09971	down
Linoleic acid	0.002	1.07298	up
Linolenic acid	0	1.13966	down
Maleamic acid	0	1.18627	up
Mandelic acid	0.003	1.05333	up
Mannitol	0	1.28916	down
Mesaconic acid	0.001	1.09989	down
Myo-inositol	0	1.13666	up
N-Acetyl-L-phenylalanine	0.002	1.06872	up
Octadecanoic acid	0	1.27733	down
9,10-Epoxyoctadecatrienoic acid	0	1.24609	up
Indomethacin	0.001	1.12263	down
Oxoglutaric acid	0	1.23002	up
Phenylalanine	0	1.26797	down
2,3-Diphosphoglyceric acid	0	1.22615	down
Proline	0	1.28514	up
Proline betaine	0.001	1.13507	down
Pyruvate	0	1.14563	down
Serine	0	1.25741	up
Threonine	0	1.24688	up
Trehalose	0	1.30756	down
Tyrosine	0	1.27859	down
Uracil	0.001	1.13457	up
Valine	0	1.14533	down

## Data Availability

The data presented in this study are available on request from the corresponding author.

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
