# Peer review of "Metabolomics Analysis of Sporulation-Associated Metabolites of Metarhizium anisopliae Based on Gas Chromatography–Mass Spectrometry"

_jof, 2023, doi:10.3390/jof9101011_

Round 1
Reviewer 1 Report
The paper is devoted to molecular dissection of an important phenomenon of fungal strain degeneration during successive long-term passaging (subculturing) in vitro (on artificial media). This process negatively affects the performance of microorganisms which serve as the basis for biological products applied in plant protection against pests, as virulence tends to diminish under these conditions. Understanding molecular mechanisms is expected to serve as a solid basis for prediction of fungal properties change and development of approaches aimed at avoiding this disadvantage of large-scale production in biotechnological industry. The authors claim that their have developed a specific analytical method for the goals of the research but it remains unclear whether this finding has already been published or they try to present it in the manuscript under review. In any case, originality of this approach is not disclosed. Reference to Fig. 1 in L100 does not seem to be relevant. Moreover, Legend to Fig. 1 is not complete as it describes only the picture to the left and not the diagram to the right.
The paper is written in comprehensive scientific language, certain polishing may be needed in some parts such as Abstract (see below). Some phrases are not very clear but this can be easily improved as discussed below
L10: the sentence lacks a predicate
L12: subculture is not a process
L17: clearly differentiated – noun is missing
L18 and further: degradation strains – do you mean degenerate strains?
L20-21: the listing of amino acid metabolism pathways is complicated and obscure: three aminoacids, then “two other”, then three amino acids – why so?
L23: are the findings mentioned published somewhere else (i.e., a previously know fact)?
L24: please be more specific when mentioning “a discernible effect”
L28-29: spore production mechanism -= this phrase is used two often here (more than once)
L78: overcapacities – what does it mean?
L79: storage pools ( … excess protein in enzymes) – does it mean that enzyme serve as a storage pool in fungal cells?
L84: check spelling of the Latin epithet
L88: why using comma between the subject and the predicate?
L88-89: infectious and saprophytic growth – while “saprophytic growth” is a natural term, “infectious growth” sounds odd, please rewrite
L105: NIST – needs a more detailed explanation
L109: comparing – maybe “by comparing”?
L116: this study will – I guess it’s not the study itself as presented in the manuscript under review but rather “continuation of this study”, “further studies in this direction” etc
L128: it is crucial to indicate the number of iterations of subcultivation, as well as duration of this process
L137: an effective means – mixture of a singular article and multiple noun
L169: please respect the punctuation
L178: this info better fits the Results section
L202: Methods only mention one “normal strain” and one “degenerate strain”; legend to the Figure 2 and text elsewhere, however, deals with “normal strains” and “degenerate strains”
L214: the former was tightly clustered – what was that?
L239: over fitting – what does it mean?
L278: two similarly colored columns – what do you mean?
L319: why only 25 pathways are shown?
L323-324: the first two amino acid metabolic pathways that exert a significant impact on sporulation are alanine, aspartate, and glutamate metabolism – the numbers of pathways do not match
Fig. 10: what CK stands for?
L374: references to metabolic pathways are needed. What “the problem pathway” is intended to mean?
L414: oxygen lipids – collocation unclear
L425: a soluble sugar, and as a soluble sugar – correct this!
L426-429: references are lacking
L454: difficult volatility – consider revision (low volatility etc)
all the impurities found are listed in the comments to he Authors
Reviewer 2 Report
The manuscript by Yang et al. analyzed the metabolic profiles of Metarhizium anisopliae based on the metabolomics approach of gas chromatography-mass spectrometry(GC-MS).
The manuscript could not be accepted in the present form due to several major concerns:
A) It is not clear what the author intended with the definition of "normal" and "degradation" strains; a more precise statement about this definition should be given.
B) None of the reported minimum standards for metabolite dereplication are accomplished for the untargeted metabolomics study. So, the authors should report the level of confidence of the 74 metabolites putatively identified in the untargeted metabolomics analysis according to the following manuscripts:
- Alseekh, S.; Aharoni, A.; Brotman, Y.; Contrepois, K.; D'Auria, J.; Ewald, J.; Ewald, J.C.; Fraser, P.D.; Giavalisco, P.; Hall, R.D. Mass spectrometry-based metabolomics: A guide for annotation, quantification and best reporting practices. Nat. Met. 2021, 18, 747–756.
- Sumner, L.W.; Amberg, A.; Barrett, D.; Beale, M.H.; Beger, R.; Daykin, C.A.; Fan, T.W.-M.; Fiehn, O.; Goodacre, R.; Griffin, J.L. Proposed minimum reporting standards for chemical analysis. Metabolomics 2007, 3, 211–221.
This manuscript should also be added to the reference list. All the identified compounds should be reported in a supplementary table.
C) In the section Data Analysis. For chemometrics analysis, it is mandatory to report the parameters of spectra pretreatment, how the data were filtered, which algorithm was used to deal with missing values and how the data were normalized for PSL-DA analysis. These data are essential to allow for reproducing the analysis. Moreover, it needs to be clarified why the PLS-DA score plot in the figure is from SIMCA and the heat map generated by Metaboanalyst. So I suggest reporting all the results using the same software, score plots and VIP score, or the results from both the software. Their concordance could strengthen the results.
Minor concerns:
-Introduction:
Lines 91-93, a reference should be added.
Lines 97-119 This is a mix of material methods and results. This is optional here. A picture is optional in the introduction. So, it should be rephrased only with the study's primary objective.
-Methods:
In line 140, the verb "was dissolved" is not appropriate better than "Were soaked.
Line 168: Name the six selected amino acids.
-Results
Section 3.3 is unnecessary, and the heat map results could be reported as a continuation of the previous section.
A better quality image seven should be reported.
-Discussion and conclusion
Figure 11 is unnecessary in the main text and could be reported as a supplementary file.
Line 427 Alternaria alternata in italic
Line 451 The reported limitation for GC-MS should be reported in the Discussion, and supporting literature should be added.
Round 2
Reviewer 2 Report
Almost all my previous criticisms have been assessed, apart from the criticism of data normalization for chemometrics analysis. Using normal data is mandatory for the multivariate statistical analysis to obtain reliable results. In section 2.4, data analysis, this information still needs to be included. How did they treat possible missing values? Did the author perform data filtering? How do the authors normalize and/or transform their data (pareto scaling? Autoscaling? Mean centring? Logarithmic Square root transformation? and so on)? This information is needed to allow for data reproducibility.
Author Response
Dear Reviewers ,
Thank you for your suggestion. As suggested by reviewer, we have add some informations in section 2.4. The contents are as follows,
For subsequent analysis, a substance should only be considered if it is present in at least 3 out of the 6 replicate samples. By default, missing values will be replaced by 1/5 of min positive values of their corresponding variables. We normalized each alignment using normalization by sum, the data were log10 transformed, and the data scaling were auto scalling (mean-centered and divided by the standard deviation of each variable.
We would like to thank the referee again for taking the time to review our manuscript.
